# Evaluation of Medication Prescribing Applications Available in Australia

**DOI:** 10.3390/pharmacy11020049

**Published:** 2023-03-06

**Authors:** Riya Amin, Melissa Cato, Sasha Rahavi, Kristin Tran, Kenneth Lee, Elton Lobo, Deanna Mill, Amy Page, Sandra Salter

**Affiliations:** 1School of Allied Health, The University of Western Australia, Crawley, WA 6009, Australia; 22479309@student.uwa.edu.au (R.A.); 22235482@student.uwa.edu.au (M.C.); 21976001@student.uwa.edu.au (S.R.); 22501529@student.uwa.edu.au (K.T.); kenneth.lee@uwa.edu.au (K.L.); deanna.mill@uwa.edu.au (D.M.); sandra.salter@uwa.edu.au (S.S.); 2Department of General Practice, The University of Melbourne, Melbourne, VIC 3010, Australia; elton.lobo@unimelb.edu.au; 3Institute for Physical Activity and Nutrition (IPAN), Deakin University, Geelong, VIC 3220, Australia

**Keywords:** medication safety, potentially inappropriate prescribing, digital health, prescribing, prescription request apps, online medications, safe prescribing, eHealth

## Abstract

Prescription request applications (apps) have changed the way consumers can obtain prescription-only medications. However, there is a lack of research surrounding such apps and their potential risks to consumers. We conducted an Australian study to (1) identify and characterise prescription request apps available in Australia and (2) assess whether prescription request apps in Australia adhere to guidelines for safe prescribing. Three online platforms (iOS App Store, Google Play store and Google search engine) were searched using 14 different search terms. Prescription request apps were identified based on pre-defined inclusion criteria. To determine whether the prescription request apps adhere to a safe prescribing framework, five medications were selected, and their corresponding consultation questionnaires were assessed against the Australian National Prescribing Service MedicineWise *12 core competencies for safe prescribing.* A total of seven prescription request apps were identified. Assessment of the prescription request apps revealed that none of the apps provided prescribers with sufficient information to meet all the competencies required for safe prescribing; rather, they inconsistently adhered to the safe prescribing framework. Thus, consumers and healthcare professionals should consider the implications and safety concerns of obtaining medications via prescription request apps.

## 1. Introduction

The growth in digital health strategies has seen an increase in the availability of eHealth web apps for consumers [1,2]. One category of eHealth apps that has emerged recently is prescription request apps. These apps allow consumers to request prescriptions for specific prescription-only medications.

The availability of medication is determined by scheduling in Australia, which is based on expert opinion with consideration to the appropriate level of health professional oversight, safety profile and need for timely access. This system allows for a balance between autonomy and medication safety. Unscheduled medications can be obtained without restrictions, while Schedules 2 (S2), 3 (S3), 4 (S4), and 8 (S8) medications have varying levels of accessibility. S2 and S3 medications are available over-the-counter (OTC) at pharmacies, although S3 medications require pharmacist approval before sale. On the other hand, prescription-only (S4 and S8) medications require a medical practitioner (or another health professional with prescribing rights) to authorise supply via providing a prescription [3]. The prescription can be authorised after interaction (consultation) between a consumer and a registered prescriber regardless of the modality of the consultation (face-to-face, telephone, video conference) [4].

It is unclear how easy it is for consumers to obtain a prescription for an S4 medication via a request on a prescription request app. Further, the role of healthcare professionals is unclear when apps are used to obtain medications. The present study seeks to (1) identify and characterise prescription request apps available in Australia and (2) assess whether prescription request apps in Australia adhere to guidelines for safe prescribing.

## 2. Materials and Methods

### 2.1. Identifying and Characterising Apps

The methods for this study are summarised in Figure 1.

#### 2.1.1. Eligibility Criteria

Apps were included if they provided the option to obtain a prescription for a self-selected prescription-only medication. Mobile and web apps were excluded if: they were unrelated to obtaining a prescription of S4 or S8 medication (such as apps that exclusively provide medication supply, medication reminder services, pill identifiers, medical dictionaries, or drug information); they were not directed for consumer use; their services were not available in Australia; or they were exclusively for animal treatment. Apps were excluded if video or verbal telehealth consultation was stated as a requirement to provide a prescription. Further, apps were excluded if there were technical issues that prevented researchers from testing the app (e.g., crashing, freezing).

#### 2.1.2. Search Strategy

Three platforms, iOS App Store, Google Play store and Google search engine, were each searched independently by two researchers to identify relevant apps. A list of primary search terms was initially identified through a panel discussion with the research team. The team comprises pharmacists, a software engineer and academics with expertise in digital health. The final search terms on web platforms (“get a prescription”, “buy pill online”, “online prescription”, “medication delivery”, “online pharmacy”, “get prescriptions online”) and mobile platforms (“prescription”, “pharmacy”, “medication script”, “medications on demand”, “medications online”, “medication delivery”, “online pharmacy”) were confirmed based on a preliminary search, where researchers reviewed the first 30 results to identify how many were relevant. The selected terms showed greater relevance for this study. The researchers used iPhone 11 and 12 devices (iOS 15.5) to search the iOS App Store, and a Google Pixel 3XL device (Android 11) was used to search the Google Play store. Personalisation was disabled while searching both mobile platforms. Google search engine was used to search for web apps, using Google Chrome while in incognito mode, to prevent data contamination from ones browsing history. All searches were conducted in July 2022.

#### 2.1.3. Databases Searched

For each search term across the 3 platforms, the first 50 results, excluding advertisements, were recorded in Microsoft Excel (Microsoft Corporation, Redmond, Washington USA; version 16.64). All results were collated, duplicates were removed, and results were screened independently by two researchers (RA and SR) according to the exclusion criteria. All iOS App Store and Google Play results were screened using the app store’s description (e.g., title, images, or text). Each Google search engine result was accessed, and the web’s homepage was reviewed to identify and exclude results that were not web apps.

The identified mobile apps were downloaded. Any discrepancies in app inclusion were resolved through discussion.

#### 2.1.4. App Feature Data Extraction

Two researchers (KT and MC) individually accessed each app and extracted information on app features in an iterative process. A standardised form was used to extract data about app features and iteratively refined as new features were identified. The data extraction form included information related to the number and type of medications available; the process for selecting a medication; prescriptions available (new, repeat, private, or subsidised by the Australian Pharmaceutical Benefits Scheme); method for receiving a prescription; and options for medication delivery (Table 1). Results from both researchers were compared, and any discrepancies were resolved through discussion. All data were extracted and recorded in Microsoft Excel (Microsoft Corporation, Redmond, WA, USA; version 16.64). Data extraction was conducted in July 2022.

### 2.2. Assessment of Safe Prescribing

#### 2.2.1. Mock Consumer Case Construction

The data extraction phase demonstrated that the apps typically use a questionnaire to facilitate the consultation and prescribing process for each medication. Two researchers (KT and MC) explored the self-select prescription function of each prescribing app to identify the questions posed when a consumer requests a prescription. Five S4 medications were specifically chosen to represent different medication safety scenarios (Table 2). Within the context of the present study, app safety refers to elements of safe prescribing and excludes safety matters concerning consumer privacy, legal jeopardy, and patient outcomes.

The questions asked by the app when selecting each medication were assessed. A mock consumer profile was constructed for each medication (Box 1) to ensure consistency of data extraction between researchers. For privacy reasons, all researchers used an alias to navigate the apps. Researchers did not purchase any prescriptions, and no government-issued identification was provided. Each app was accessed between 1 and 12 August 2022.

Box 1Mock consumer patient profile characteristics used to access each appTo ensure consistency between researchers, questions were extracted from the apps using standardised mock consumer profiles. In total, five consumer profiles were created (one for each drug). The general characteristics for each consumer profile were the same and included the following characteristics: 32 years old; body mass index (BMI) of 20; no medical conditions (except for the specific condition the selected medication was being prescribed for); was not using any other medications (including herbal, complementary, or over-the-counter medications); non-smoker, did not consume alcohol or use recreational drugs; and had no allergies. When requesting a prescription for sildenafil, the mock consumer was assigned as a male who was not transgender. When requesting each of the other four medications, the mock consumer was assigned as a female who was not transgender and was not pregnant or breastfeeding.

#### 2.2.2. Assessment of Adherence to NPS Competency Guidelines

Following a review of the literature, the Australian National Prescribing Service (NPS) MedicineWise *12 core competencies for safe prescribing* was chosen to examine the prescribing process [13]. This framework was selected as it provides a guideline for the safe and effective prescribing of medications. It comprises 12 competencies which are divided into the following four stages of prescribing: information gathering, decision making, communicating the decision, and monitoring and review (Table 3) [13]. To determine whether prescription request apps in Australia adhere to guidelines for safe prescribing, the questions asked by each app were mapped to the competency that they addressed for each of the five chosen medications. The lists of criteria were developed by two researchers (KT and MC). To minimize bias, each app was then assessed independently by two other researchers (RA and SR), results were compared, and any discrepancies were resolved through discussion. Finally, competency 11 was excluded from this study as it relates to communication within a hospital setting and the identified apps are designed for use in the community.

Competencies 3–7 and 10: The app was considered to satisfy a competency if the questionnaire included at least one question relevant to that competency. For example, when requesting fluticasone/salmeterol, if the app asked about the consumer’s frequency of reliever use, competency 6 (disease management) was deemed met as the app enquired about the management of the consumer’s respiratory disease and how effective their current fluticasone/salmeterol treatment regimen is. 

Competencies 1, 2, 8, 9 and 12: These competencies were broad and not reasonably able to be met with a single question (e.g., an accurate medical history should explore the consumer’s age, gender, medical conditions, known allergies, pregnancy and breastfeeding status, body mass index, smoking history, and alcohol consumption). As such, we developed separate lists of criteria specific to each of the five chosen medications to represent safe prescribing for each medication for competencies 1, 2, 8, 9 and 12 (Appendix A). These competencies were considered to have been met if the app satisfied at least 50% of the developed criteria for each medication; this 50% criterion was determined by the team given that 50% in assessments typically constitutes a ‘pass’ and to facilitate consistency in decision making. The criteria for competency 1 (medical history) and competency 2 (medication history) were derived from the Royal Australian College of General Practitioners *Standards for general practices*, criterion QI2.1A and QI2.1B [14]. Competency 8 encompassed two concepts of safe prescribing and hence, was examined in two parts: questions about whether the consumer was using *other treatment* and if the consumer had any *contraindications* for the requested medication. Criteria for competencies 8 (contraindications) and 9 (dose regimen) were derived from the standard QUM decision-making tools required by pharmacists and doctors [15], including the Australian Medicines Handbook (AMH), AusDI, Monthly Index of Medical Specialties (MIMS), Stockley’s Drug Interactions, and the Therapeutic Guidelines [8,16,17,18,19,20,21].

### 2.3. Data Analysis and Synthesis

Individual apps were anonymised using an alphabetical indicator (A-G). The data from each individual app were then tabulated against the individual assessment outcomes. Binary (present/absent) responses were indicated, as well as continuous data or qualitative descriptions in the tables.

## 3. Results

### 3.1. Application Characteristics

#### 3.1.1. Search and Categorisation

Across the three platforms, a total of 1134 search results were identified (n = 357 from Apple Store, n = 392 from Google Play Store, and n = 385 from Google search engine) (Figure 2). Of these, 390 were duplicates and therefore were removed. Further, 659 results were excluded as they were neither medication nor prescription related, unavailable in Australia, not consumer-directed, or not an app. The remaining 85 results were screened against the set criteria. A total of seven apps were identified to meet the set criteria.

#### 3.1.2. Prescribing App Features

A total of seven apps were categorised as prescription request apps (Table 4). Six of the apps allowed consumers to request prescriptions for S4 medications for various medical conditions, whereas one app only offered oral contraceptives. All apps offered medication delivery when a prescription was requested.

Three apps (Apps D, F and G) exclusively deliver the requested medication and do not issue the prescription directly to the consumer or a nominated pharmacy. All seven apps provided prescriptions for medications that the consumer had previously used, whilst Apps A, B, D, E, F and G also provided the option to request new prescription medications. 

Four apps (A, B, D and G) allowed prescription requests (by brand or generic) from a list using free text. App E prompted consumers to first select the medical condition requiring treatment, then use a search function to select a medication (using brand or generic name) from a pre-specified list. Where pre-specified medications were not listed, consumers could type any word in the search box and, in this instance, were directed to a digital medical consultation. App E asked medical condition-specific questions rather than medication-specific questions. To search for medications on App C, consumers must type at least three characters for the requested item to appear. None of the apps prescribed S8 medications.

### 3.2. Analysis of Prescription Request Apps

When requesting each of the five drugs, the seven apps asked between 7 and 29 questions. The average number of questions asked across all five drugs on each app was: 11 (App A), 21 (App B), 12 (App C), 14 (App D), 11 (App E), and 18 (App F). App G asked the most questions (n = 29) for Levlen^®^. However, this app only prescribed oral contraceptives.

#### Adherence to NPS Competency Framework for Safe Prescribing

None of the apps met all competencies for any medication explored in this study. Apps B (when prescribing fluticasone/salmeterol and colchicine) and F (when prescribing Levlen^®^) were the most adherent to the NPS framework and met eight of the competencies, as shown in Table 5. Table 5 illustrates the gaps identified when prescription request apps were assessed against the four stages and the 11 core competencies for safe prescribing.

##### Prescribing Stage 1: Information Gathering

Five of the seven apps satisfied the competencies concerned with reviewing medical history and medication history. This medication and medical review process included a standardised health questionnaire when each medication was selected. Apps A and E did not utilise this standardised approach to gather health information and demonstrated gaps in competencies 1 and 2. None of the apps addressed competency 3, which involved undertaking a physical examination or further investigations. However, all the apps asked consumers about a prior diagnostic test (e.g., blood test, blood pressure measurement, pap smears, cervical screening, etc.) for at least one medication. Competency 4 (adherence) was not addressed by any app except when investigating colchicine on App C.

##### Stage 2: Decision Making

Competency 5, which encapsulates shared decision-making, was not addressed by any app. Competencies 6 and 7 were addressed by Apps F and G across all medications that the apps offered. The remaining apps inconsistently addressed these competencies. Competency 8, contraindications, was met by most of the apps, however enquiring about other treatments was not met by Apps A, D, E or G. Competency 9, which concerned dosing regimen, was inconsistently assessed across the apps excluding App B, which asked about dose regimen for all five medications.

##### Stage 3: Communicate Decision

Competency 10 (communicating prescribing decisions) was carried out by one app, App A. This app provided the consumer with the option for the app-based prescriber to share the result of the consultation with their regular general practitioner (GP). Consumers can search for their regular GP by entering the GP’s name or the medical centre. App A also asked the consumer to indicate if they ‘understood everything’ from the questionnaire, with a ‘yes’ or ‘no’ response. If ‘no’ was selected, the consumer was directed to request a telehealth consultation. The consumer could then proceed to telehealth or toggle back to ‘yes’. Selecting ‘yes’ progresses to asking for the preferred ‘collection method’ for the medication.

##### Stage 4: Monitor and Review

Competency 12, treatment monitoring, was consistently addressed by four apps. The apps monitored treatment by reviewing the therapeutic and/or adverse effects of treatment through the questionnaire process.

## 4. Discussion

To our knowledge, this is the first study to comprehensively examine consumer-directed prescription request apps. As of August 2022, there are seven prescription request apps in Australia. The assessment of prescription request apps revealed gaps in the process of safe prescribing, as defined by the NPS MedicineWise prescribing competencies framework [13].

Five apps utilised a standardised general health questionnaire to gather information. This ensured that important consumer demographic questions were consistently asked each time a prescription was requested. This contrasts with traditional health consultations, where information gathering may vary depending on the patient or doctor, and changes to patient information between GP visits may not be recognised. While the app-based standardised questionnaire provides consistency in information gathering, there is potential for misuse. It is possible that consumers could recognise the answers that are required to enable prescribing and choose to answer questions dishonestly in order to obtain their desired medication. While this may also occur in face-to-face consultations, in-person consultations allow finessed questioning, whereas it is not possible to probe answers given in a questionnaire, so the genuine need is almost impossible to determine via the app. Moreover, the two apps did not utilise a standardised general health questionnaire and therefore did not have a comprehensive or consistent information-gathering process. This is concerning as the quality use of medicines relies on prescribers being aware of patients’ general health information in order to prescribe medications that are appropriate and safe.

An important aspect of prescriber consultations is the process of shared decision-making [13]. This encompasses a collaboration between the prescriber and the consumer to discuss and agree on appropriate treatment options in partnership [13]. In the case of ongoing therapy, consumer request potentially prevents the optimisation of the consumer’s treatment plan and may affect clinical outcomes [13]. The fundamental process of shared decision-making requires prescribers to utilise information regarding *indication, disease management, other treatment*, and *contraindications* to form a clinical decision surrounding a consumer’s therapy [13]. While some of the questionnaires satisfied these relevant competencies, it was evident that the model of prescription request apps focused on identifying consumers who are *not* suited for the requested medication, rather than selecting the most optimal treatment for the consumer’s condition. This is reflected by the emphasis the apps have for assessing contraindications instead of approaching therapy in a more holistic manner.

Communication of accurate and complete prescribing information to both the patient and other HCPs is critical to achieving optimal patient care. The apps did not have an interactive communication facility; rather, they were based on screening questions. One app asked the consumer to confirm their understanding of the consultation based on a ‘yes’ or ‘no’ response to a ‘do you understand everything’ question. If ‘yes’ is selected, the prescription request proceeds to the ‘collection method’ for the medication (collect from a pharmacy, provide e-script, or deliver to door). If ‘no’ is selected, the consumer is directed to request a telehealth consultation. Consumers can toggle back to ‘yes’ (and medication collection options) if they decide not to proceed to telehealth. No further information is provided by the app. Seemingly this addresses a communication parameter, yet practically it poses problems: consumers have already chosen their medication and will either answer the question rhetorically (yes) to satisfy their goal of obtaining the prescription or change their answer as needed to proceed to medication delivery. Communicating via questionnaire is one-sided. Thus, prescription request apps lack the ability to engage in interactive communication with the consumer, which can lead to medication and prescribing errors.

### 4.1. Strengths & Limitations

Our study has notable limitations that are important to discuss. No prescriptions were obtained as the researchers could not procure the medications using the identification required by the apps. This meant that we were unable to assess what happens after responses to the questionnaires are submitted for review by the health professional; as such, our study was not able to determine whether a consumer’s request may lead to a referral to a consultation with a health professional, and therefore we cannot definitively conclude the safety of procuring medications from an app. Further, we selected only five medications advertised on the apps and used a standardized consumer to represent a cross-section of considerations that must be made when prescribing and examined the prescribing process in detail. However, the prescription request apps may have performed differently had we chosen different medications or consumer parameters. Similarly, considering the selective nature of this study, the apps may have performed differently had we chosen to assess all advertised medications. Further, the mock consumer profile that we created in Box 1 depicts consumers who are healthy and do not have co-morbidities; thus, we did not determine whether creating a mock consumer profile with significant co-morbidities would have yielded different results. Finally, there does not appear to be a unified framework for assessing prescribing competencies; as such, there may be more appropriate frameworks than the NPS 12-core competencies we have used. However, our study highlights some potential gaps in the identified apps. Despite these limitations, to our knowledge, this is the first study to identify prescription request apps within Australia. This study is also the first to comprehensively assess prescription request apps, with the overarching goal of determining if the apps safely prescribe medications. Additionally, as apps continue to emerge, this study may provide a framework for ongoing and further examination of medication-related apps, as well as highlight the potential pitfalls of these unregulated apps in the medication prescribing sphere.

### 4.2. Future Research

There are no known studies investigating the quality and appropriateness of prescribing services offered by prescription request apps in parallel to traditional face-to-face doctor consultations. Additionally, as we only focused on prescription request apps that do not involve video/verbal telehealth consultations, it is unknown how apps that utilize this functionality would fare against prescribing competencies. Future research could consider, within the bounds of the law, utilising simulated patients to obtain prescriptions to thoroughly assess the entirety of the prescribing process, and compare this to standard care. The evolving nature of telehealth may also influence consumer perceptions and willingness to use prescription request apps, and experiential research is needed to examine consumer drivers surrounding their use.

## 5. Conclusions

Prescription request apps may provide a convenient and accessible approach to health care. However, this method of obtaining medication diminishes healthcare professional involvement, which can impact patient care. This study provided an overview of the features and characteristics of currently available prescription request apps in Australia and the adherence of prescription request apps to current prescribing guidelines. It has important policy considerations for medicines regulation (scheduling) and health professionals (prescribers and pharmacists). It is imperative that the prescription request apps are monitored to ensure that their convenience does not come at the cost of lower-quality healthcare for the consumer. Legal, ethical and privacy issues must be examined in future, with a recommendation that prescription request apps are regulated to ensure appropriate standards of health care and medication safety are met.

## Figures and Tables

**Figure 1 pharmacy-11-00049-f001:**
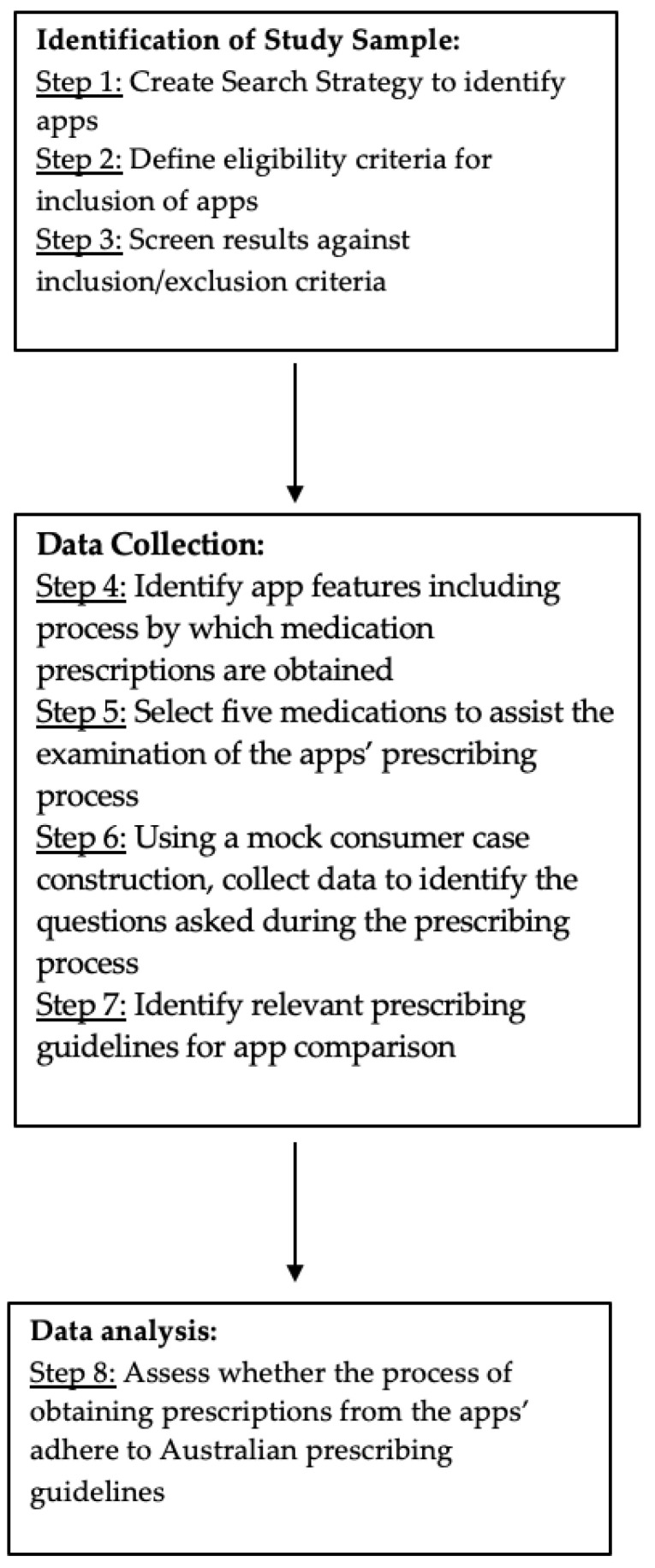
Flow chart of the study methods.

**Figure 2 pharmacy-11-00049-f002:**
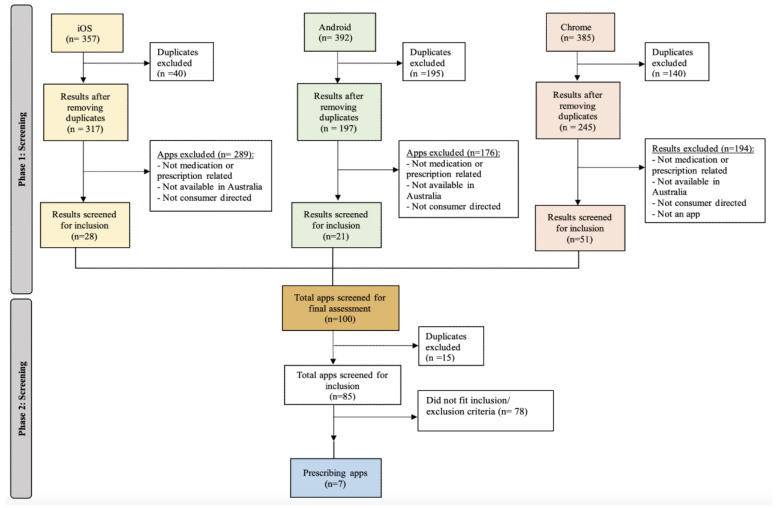
Flowchart illustrating app identification.

**Table 1 pharmacy-11-00049-t001:** Data extraction form.

Item	Response
App Names	Free text
Process of obtaining a script	Free text
Method for receiving prescription	Free text
Medicine Catalogue	Free text
Dispensed by a local pharmacy	Free text
Who can you obtain a script for?	Anyone/Yourself. Elaborate if necessary
Prescription type	New and/or Repeat prescriptions Private and/or subsidised by the Australian Pharmaceutical Benefits Scheme
Are brands available for purchase?	Yes/No
Bulk Billing	Yes/No
Personal Information Collected	Free text
Popularity of app/number of downloads	Free text
Consult style (e.g., online, telehealth)	Free text
Cost for online consult	Free text
Other comments	Free text

**Table 2 pharmacy-11-00049-t002:** Five medications selected to assess the questions asked by each prescribing app and the rationale for selecting these medications.

*Medication*	*Rationale for Selection*
*Combined oral contraceptive pill (Levlen^®^, containing ethinylestradiol 30 mcg and levonorgestrel 150 mcg)*	Appropriate contraceptive choice requires careful consideration of drug interactions, contraindications, adverse effects, and an individual’s circumstances [5].
*Fluticasone/salmeterol MDI*	Misused inhaler devices can impact disease management; asthma requires frequent monitoring [6].
*Sildenafil*	Potential for serious drug interactions; commonly self-selected, which may lead to misdiagnosis of other serious conditions such as cardiovascular disease [7,8].
*Sertraline*	A commonly prescribed antidepressant in Australia [9]. The prevalence of poor mental health conditions may lead to inappropriate self-medicating [10,11].
Colchicine	Has a narrow therapeutic window, the potential for serious drug interactions and can have complex dosing regimens [12].

**Table 3 pharmacy-11-00049-t003:** The NPS MedicineWise *12 core competencies for safe prescribing* framework can be found at: https://www.nps.org.au/australian-prescriber/articles/the-competent-prescriber-12-core-competencies-for-safe-prescribing (accessed on 12 August 2022). This framework guided the assessment of safe prescribing across the prescribing apps. The questions asked to obtain the medications selected in this article were mapped to the competencies they addressed.

Four Stages of Prescribing	12 Core Competencies
(1) Information gathering (skill of gathering relevant information to inform selection of treatment)	1. Take and/or review medical history
	2. Take and/or review medication history and reconcile this with medical history
	3. Undertake further physical examination/investigations where appropriate
	4. Assess adherence to current and past medication and risk factors for non-adherence
(2) Decision-making (collaborative decision-making with the patient/carer; selection of treatment)	5. Identify key health and/or medication-related issues with the patient, including making or reviewing the diagnosis
	6. Determine how well disease and symptoms are managed/controlled
	7. Determine whether current symptoms are modifiable by symptomatic treatment or disease modifying treatment
	8. Consider ideal therapy (drug and non-drug), taking into account actual and potential contraindications/concerns: drug–patient, drug–disease, drug–drug interactions
	9. Select drug, form, route, dose, frequency, duration of treatment
(3) Communicate decision (safely and effectively communicate treatment decisions to other health professionals and the patient/carer in both the ambulatory and the inpatient setting)	10. Communicate prescribing decision in an ambulatory care setting
	11. Communicate prescribing decision in an inpatient setting
(4) Monitor and review (review the therapeutic and adverse impact of treatment)	12. Review control of symptoms and signs, adherence, and patient’s outcomes

**Table 4 pharmacy-11-00049-t004:** Features of prescribing request apps available in Australia.

	App A	App B	App C	App D	App E	App F	App G
Platforms	Web, iOS, Android	Web, iOS, Android	Web, iOS, Android	Web	Web, iOS, Android	Web	Web
Telehealth Option	✓	✓	✓		✓		
Who Can You Obtain a Prescription For?							
Self	✓	✓	✓	✓	✓	✓	✓
Others	✓		✓		✓		✓
Prescription Available							
New (without telehealth)	✓	✓			✓	✓	✓
Repeat (without telehealth)	✓	✓	✓	✓	✓	✓	✓
PBS	✓		✓	✓			
Private	✓	✓	✓	✓	✓	✓	✓
Method of Obtaining Prescription							
E-script	✓		✓		✓		
Postal		✓		*	✓	*	*
Sent to local pharmacy	✓		✓				
Number of Medications Available	184	190	-	76	-	-	11
Available Medicine Catalogue	Heartburn, anaphylaxis, blood pressure, blood thinners, cholesterol, diabetes, eye condition, gout, headache & migraines, men’s health, mental health, neurological diseases, opioid safety, pain management, respiratory conditions, sexual health, skin care, sleep, thyroid medications, travel, urinary incontinence, vitamins, weight loss, women’s health	Heartburn, allergies, blood pressure, cholesterol, cold & flu, diabetes, gout, headache & migraines, men’s health, mental health, respiratory conditions, skin care, women’s health	Heartburn, allergies, blood pressure, bone health, cholesterol, diabetes, gout, headache & migraines, men’s health, mental health, pain management, respiratory conditions, skin care, urinary incontinence, women’s health	Heartburn, cold & flu, gout, headache & migraines, mental health, respiratory conditions, sexual health, skin care, sleep, smoking cessation thyroid medications, travel, weight loss, women’s health	Heartburn, allergies, blood pressure, cholesterol, cold & flu, diabetes, gout, headache & migraines, men’s health, mental health, respiratory conditions, skin care, women’s health	Heartburn, cold and flu, respiratory conditions, skin care, sleep, women’s health	Women’s health (contraception only)
**Methods to Obtain Prescription**	Search medication by name or medication catalogue followed by selecting a medication, complete an online questionnaire*, select method for receiving prescription doctor reviews information (*Some medication requires telehealth)	Select medical condition, type in medication of desire with dose, complete an online questionnaire, select collection method, doctor reviews information	Select ‘new prescription’ or ‘repeat prescription’ tab. Telehealth consult is required for all new prescriptions. For repeat prescription - search medication by name, select collection method, complete an online questionnaire, reviewed by doctor	Search for medication by name or search for medication catalogue followed by selecting a medication, complete an online questionnaire, reviewed by doctor, medication will be sent to the patient	Select medical condition, type in medication of desire with dose, complete an online questionnaire, select collection method, doctor reviews information	Select medical conditions, fill in personal details, complete an online questionnaire, reviewed by doctor (receive text or email when script is ready)	Select ‘contraception’ tab, complete a questionnaire, select pill brand, reviewed by doctor, medication sent to the patient

**Table 5 pharmacy-11-00049-t005:** Assessment of apps against the NPS MedicineWise 12 core competencies for safe prescribing.

Four Prescribing Stages	Core Competencies		App A					App B					App C					App D					App E					App F					App G				
Information gathering	1	Medical history		✓				✓	✓	✓	✓	✓	✓	✓	✓		✓	✓		✓		✓	✓	✓		✓	✓	✓	✓				✓				
	2	Medication history						✓	✓	✓	✓	✓	✓	✓	✓		✓	✓		✓		✓						✓	✓				✓				
	3	Further information																																			
	4	Adherence															✓																				
Decision-making	5	Shared decision-making																																			
	6	Disease management	✓	✓		✓		✓	✓		✓	✓	✓		✓			✓		✓			✓					✓	✓				✓				
	7	Indication				✓	✓	✓	✓		✓	✓		✓	✓					✓		✓		✓				✓	✓				✓				
	8a	Other treatment		✓					✓		✓	✓		✓			✓					✓				✓		✓	✓								
	8b	Contraindications	✓	*		✓	✓	✓	*	✓		✓	✓	*			✓	✓		✓		✓	✓	*	✓	✓	✓	✓	*				✓				
	9	Dose regimen	✓	✓				✓	✓	✓	✓	✓	✓	✓			✓	✓					✓	✓			✓	✓					✓				
Communicate decision	10	In an outpatient setting	✓	✓	✓	✓	✓				✓																										
Monitor and review	12	Treatment monitoring		✓		✓		✓	✓	✓	✓	✓	✓	✓	✓		✓	✓				✓	✓	✓	✓	✓	✓	✓	✓				✓				
Total number of questions			13	7	9	18	7	24	21	22	17	19	17	8	12		9	12		20		10	15	7	12	14	8	21	14				29				
			Ethinylestradiol/levonorgestrel						Fluticasone/Salmeterol						Sildenafil							Colchicine				Sertraline				Not available in-app							

Core competencies are written as main themes, and (Comp.) represents the competency the theme relates to. For full competency wording, see NPS 12 core competencies; * No contraindications were identified in the quality use of medicine decision-making tools (Australian Medicines Handbook, AusDI, Monthly Index of Medical Specialties, Stockley’s Drug Interactions, and Therapeutic Guidelines); ✓ = competency met (for competencies 1, 2, 8, and 9 the apps needed to satisfy at least 50% of the developed criteria to meet the overall competency).

## Data Availability

The data presented in this study is available in the Appendix A.

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
