# Peer review of "Evaluation of Medication Prescribing Applications Available in Australia"

_pharmacy, 2023, doi:10.3390/pharmacy11020049_

Round 1

Reviewer 1 Report (Previous Reviewer 3)

To the authors,

Congratulations on your research article. Your article has greatly improved after the last revision. Below are a few additional comments and suggestions:

·       Figure 1: Please recalculate 'Android – apps excluded' should be n=176 (197-21=176) instead of 174

·       Line 162: “Further, 682 results were excluded ..” Please recalculate, should be 659 (289+176 (see above) +194 = 659) instead of 682

·       Line 12 (page 10): “… then type in the medication name and strength that the consumer is requesting.” Please consider explaining whether a specific brand of medicine can be chosen or only the active pharmaceutical ingredient.

·       Line 100 (page 12): “one app asked the consumer to confirm their understanding of the consultation based on a ‘yes’ or ‘no’ response to a ‘do you understand everything’ question.”  Please indicate what happens if "no" is indicated. In this case, will further information be provided by the app?

·       Reference 15: Please add the ‘accessed at’ date

·       Reference 17: What does AusDi stand for? Please consider adding additional information.

Author Response

Figure 1: Please recalculate 'Android – apps excluded' should be n=176 (197-21=176) instead of 174

Done

Line 162: “Further, 682 results were excluded ..” Please recalculate, should be 659 (289+176 (see above) +194 = 659) instead of 682

Done.

Line 12 (page 10): “… then type in the medication name and strength that the consumer is requesting.” Please consider explaining whether a specific brand of medicine can be chosen or only the active pharmaceutical ingredient.

Thank you, this has been reworded as shown:

App E prompted consumers to first select the medical condition requiring treatment, then use a search function to select a medication (using brand or generic name) from a pre-specified list. Where pre-specified medications were not listed, consumers could type any word in the search box, and in this instance were directed to a digital medical consultation.

Line 100 (page 12): “one app asked the consumer to confirm their understanding of the consultation based on a ‘yes’ or ‘no’ response to a ‘do you understand everything’ question.”  Please indicate what happens if "no" is indicated. In this case, will further information be provided by the app?

Thank you for the query. This has been reworded as shown:

If ‘no’ was selected, the consumer was directed to request a telehealth consultation. The consumer could then proceed to telehealth, or toggle back to ‘yes’.  Selecting ‘yes’ progresses to asking for the preferred ‘collection method’ for the medicine.

Reference 15: Please add the ‘accessed at’ date

Done.

Reference 17: What does AusDi stand for? Please consider adding additional information.

AusDI is the name of the resource.  It does not stand for anything.

Reviewer 2 Report (New Reviewer)

Here these are my comments:

Line 13, you should state which research design you have used. Is this a cross-sectional study?

In the introduction section, you say nothing on the interconnection of this study and patient safety specifically medication safety. Try to improve this section with more literature on medication safety both in Austrlia and internationally.

Material and methods should be started with the identification of a research design. You are advised to use the appropriate Equator checklist to present your article. Here this is the link: EQUATOR Network | Enhancing the QUAlity and Transparency Of Health Research (equator-network.org)

Please attach a copy of the data extraction tool as that starndard tool under 2.1.4. as supplementary file.

Elaborate on the process of data extraction and comparison of apps with the mentioned standards. How bias was prevented?

Add ethical considerations in this research.

Add the data analysis and research synthesis process before the results.

In the conclusion, mention the practical implications of your research for policy making, education and management in healthcare. 

Author Response

Line 13, you should state which research design you have used. Is this a cross-sectional study?

Thank you for the suggestion.  We have decided not to state a research design as the method is novel and does not neatly fit a study design.  It uses components of a cross-sectional study as it is undertaken as a snapshot in time without an intervention but there are no human participants.  It uses components of a simulated patient study but again it does not have any human participants. It uses components of systematic review and audit, but the search is novel as it is required to identify the apps themselves (as opposed to searching databases to find the literature about them). We have decided that it is appropriate to leave the design unnamed.

In the introduction section, you say nothing on the interconnection of this study and patient safety specifically medication safety. Try to improve this section with more literature on medication safety both in Austrlia and internationally.

Thank you for the suggestion. We have added:

The availability of medication is determined by scheduling in Australia based on expert opinion with consideration to the appropriate level of health professional oversight, safety profile and need for timely access. This system allows for a balance between autonomy and medication safety. 

Material and methods should be started with the identification of a research design. You are advised to use the appropriate Equator checklist to present your article. Here this is the link: EQUATOR Network | Enhancing the QUAlity and Transparency Of Health Research (equator-network.org)

We appreciate the thoughtful suggestion and helpful provision of the information about the EQUATOR Network.  We have decided not to incorporate this suggestion as there is no checklist that appropriately matches our research method (as above).

Please attach a copy of the data extraction tool as that starndard tool under 2.1.4. as supplementary file.

Thank you for this prompt.  We apologise for not having done so already.  We have provided the tool as a supplementary file.

Elaborate on the process of data extraction and comparison of apps with the mentioned standards. How bias was prevented?

Thank you.  This information has been described in the manuscript now:

The lists of criteria were developed by two researchers, and to minimise bias each app was then assessed independently by two other researchers, results compared and any discrepancies resolved through discussion.

Add ethical considerations in this research.

We have no added ethical considerations as there were no human participants so ethical approval was not necessary.  This is akin to publishing a literature review where no ethical approval is required.

Add the data analysis and research synthesis process before the results.

Thank you.  We have done this now.  The section reads:

Data analysis and synthesis

Individual apps were anonymised using a numerical indicator (1-7). The data for each individual app were then tabulated against the individual assessment outcomes. Binary (present/absent) responses were indicated, as well as continuous data or qualitative descriptions in the tables.

In the conclusion, mention the practical implications of your research for policy making, education and management in healthcare. 

Thank you.  We have elaborated on this information now.

It has important policy considerations for medicines regulation (scheduling) and health professionals (prescribers and pharmacists). It is imperative that the prescription request apps are monitored to ensure that their convenience does not come at the cost of lower-quality healthcare for the consumer. Legal, ethical and privacy issues must be examined in future, with a recommendation that prescription request apps are regulated to ensure appropriate standards of health care and medication safety are met.

Round 2

Reviewer 2 Report (New Reviewer)

Dear Authors,

It does not make sense to conduct a research and do not specify a method type for it. There must be a description of the research design that allows readers to evaluate the quality of your research and even others repeat it. This is the characteristic of publication in a scientific journal. 

Sincerely,

Author Response

We have added a line at the start of the methods to specify a research design.  It now reads:

We undertook a cross-sectional study of the prescription request apps that were available in Australia in July and August 2022.

This manuscript is a resubmission of an earlier submission. The following is a list of the peer review reports and author responses from that submission.

Round 1

Reviewer 1 Report

Interesting topic and study approach. I appreciate the detailed data collected and commend the group for undertaking this work.

After reading the title, abstract and introduction, I had several questions...some of which were answered in the body of the paper. I suggest making the following elements clearer up front:

- that your group looked only at the "safe prescribing" elements of app safety (not consumer privacy or legal jeopardy or patient outcomes which could also be considered "safety")

- you don't address the legal status of any of these apps or if/how they're currently subject to any regulatory control in Australia or elsewhere. For example what prescriber's name appears on the label from any of the prescribing apps? Table 5 indicates that a doctor reviews questionnaire information...in some jurisdictions, a specific physician must be associated with an online prescribing service in order for a MD name to appear on the Rx label and that individual would carry the legal liability for patient assessment, follow-up etc.  Since your study doesn't address this at all, it would be clearer if your assumptions and current Australian legal framework were briefly mentioned in the introduction (i.e. if all these apps are illegal but consumers use them anyway, that would be good to know). If a physician IS associated with any of the prescribing apps, how are they (the MDs) regulated? This should be explained briefly, even if it's under review or controversial. Is there no opportunity for a patient to e-mail the prescriber for further information? Is there a College of Physicians policy on this practice? If so, this should be included.

- Abstract should include the clear statement that none of the prescribing apps fully adhere to the safe prescribing competencies (needs rewording). The problems with this element seemed to be understated in the abstract relative to the data provided. If there are MDs or other qualified prescribers affiliated with any of these apps, aren't they, in fact, the ones who aren't adhering to safe prescribing guidelines (not the inanimate apps? Unless this is handled very differently in Australia than in other jurisdictions or wherever the headquarters of the apps are, if located overseas).

Methods: a more fulsome explanation of exclusion criteria...were video  telemedicine consults included as verbal consultation? Also, it would be helpful to include an example of a drug from each schedule for international readers.

Table 3 - good choices of drugs and rationale. Perhaps add details of generic names for Levlen ingredients (for international readers who may not recognize this name).

Tables 4 & 5 - indicate country of origin of the apps, if available. If all are headquartered in Australia, please indicate this.

Conclusion: I suggest that you make it clear that prescribing from these apps wasn't found to meet current guidelines (not that you explored the quality of safe prescribing). Also it would be good to know who you feel is currently responsible for this situation, i.e. are consumers using an illegal service, are the services originating overseas or is it in a legal grey area where MDs aren't practising to their usual expected standard? When you recommend monitoring and regulation - who would regulate them? Are Australian health professionals working for such companies not subject to the same regulations as other professionals working in traditional settings? This information will be important for international readers who may be familiar with similar apps from a very different context.

Author Response

Dear Reviewer,

Thank you for taking the time to critically review our manuscript and for providing constructive feedback. In addition to addressing your comments and updating our manuscript accordingly, we have also re-read the entire manuscript (and amended where appropriate) to improve clarity.

  1. That your group looked only at the “safe prescribing” elements of app safety (not consumer privacy or legal jeopardy or patient outcomes which could also be considered “safety”)

We agree with this comment and how our definition of safety is only considered in the context of safe prescribing. To demonstrate this, we have included the following sentence in our abstract and edited line 64-65 of our introduction.

“In the context of this study, app safety refers to elements of safe prescribing and excludes safety matters concerning consumer privacy, legal jeopardy and patient outcomes.”

  1. You don't address the legal status of any of these apps or if/how they're currently subject to any regulatory control in Australia or elsewhere. For example what prescriber's name appears on the label from any of the prescribing apps? Table 5 indicates that a doctor reviews questionnaire information...in some jurisdictions, a specific physician must be associated with an online prescribing service in order for a MD name to appear on the Rx label andthatindividual would carry the legal liability for patient assessment, follow-up etc.  Since your study doesn't address this at all, it would be clearer if your assumptions and current Australian legal framework were briefly mentioned in the introduction (i.e. if all these apps are illegal but consumers use them anyway, that would be good to know). If a physician IS associated with any of the prescribing apps, how are they (the MDs) regulated? This should be explained briefly, even if it's under review or controversial. Is there no opportunity for a patient to e-mail the prescriber for further information? Is there a College of Physicians policy on this practice? If so, this should be included.

We agree, and have now updated the introduction to provide the legal context in Australia.

  1. Abstract should include the clear statement that none of the prescribing apps fully adhere to the safe prescribing competencies (needs rewording). The problems with this element seemed to be understated in the abstract relative to the data provided. If there are MDs or other qualified prescribers affiliated with any of these apps, aren't they, in fact, the ones who aren't adhering to safe prescribing guidelines (not the inanimate apps? Unless this is handled very differently in Australia than in other jurisdictions or wherever the headquarters of the apps are, if located overseas).

That is correct.  We have clarified the abstract with this information.

  1. Methods: a more fulsome explanation of exclusion criteria...were video telemedicine consults included as verbal consultation? Also, it would be helpful to include an example of a drug from each schedule for international readers.

We have clarified the exclusion criteria in Table 2 as well as clarified our operational definition. We have also added examples of drugs for each schedule.

  1. Table 3 - good choices of drugs and rationale. Perhaps add details of generic names for Levlen ingredients (for international readers who may not recognize this name).

In response to this comment, we have included the names of the active ingredients in Levlen®. This is located in Table 3.

  1. Tables 4 & 5 - indicate country of origin of the apps, if available. If all are headquartered in Australia, please indicate this.

We have actioned this comment by including the apps’ country of origin in Table 4 and 5.

  1. Conclusion: I suggest that you make it clear that prescribing from these apps wasn't found to meet current guidelines (not that you explored the quality of safe prescribing). Also it would be good to know who you feel is currently responsible for this situation, i.e. are consumers using an illegal service, are the services originating overseas or is it in a legal grey area where MDs aren't practising to their usual expected standard? When you recommend monitoring and regulation - who would regulate them? Are Australian health professionals working for such companies not subject to the same regulations as other professionals working in traditional settings? This information will be important for international readers who may be familiar with similar apps from a very different context.

Thank you for this feedback. We have amended the conclusion accordingly.

Reviewer 2 Report

This review is a timely, well defined, and well documented review of an understudied interaction between patients and methods to access care. The criteria for review were based on standards.

While a helpful review of the inclusion and exclusion of the standards are defined in the text, the prose for inclusion can be difficult to follow. One consideration would be to add further details to the figures to add to the content, such as Table S1. 

The discussion offers liberty for editorial considerations. Line 395 reads "App prescribers cannot possibly achieve a comprehensive understanding of how the consumer is using the medication, and whether a prescribed dose is appropriate." The sentiment is understood. This statement may not stand objectively. 

The discussion of the Stages in context of the results help align the results and the standards in the review of apps. These abstract considerations would benefit from details either specific to the apps or in the user experience with the apps. Ex. - Line 434,  "Prescribing apps did not review  any other aspect of the consumer’s therapy, such as disease progression or resolution". What would be an expected component of "Monitor & Review" that these apps did not address? 

Author Response

Dear Reviewer,

Thank you for taking the time to critically review our manuscript. We have considered your comments and have provided a response below, as well as made relevant revisions to our original manuscript.

  1. While a helpful review of the inclusion and exclusion of the standards are defined in the text, the prose for inclusion can be difficult to follow. One consideration would be to add further details to the figures to add to the content, such as Table S1. 

Thank you for your comment. We have expanded the content in the figure legends of the supplementary material, to make it easier for readers to follow.

  1. The discussion offers liberty for editorial considerations. Line 395 reads "App prescribers cannot possibly achieve a comprehensive understanding of how the consumer is using the medication, and whether a prescribed dose is appropriate." The sentiment is understood. This statement may not stand objectively. 

Thank you. We agree that our original statement was subjective and have therefore revised it accordingly in our manuscript.

  1. The discussion of the Stages in context of the results help align the results and the standards in the review of apps. These abstract considerations would benefit from details either specific to the apps or in the user experience with the apps. Ex. - Line 434, "Prescribing apps did not review any other aspect of the consumer’s therapy, such as disease progression or resolution". What would be an expected component of "Monitor & Review" that these apps did not address?

Thank you for your comment. We have provided additional clarification to the example you have provided, and re-read our discussion to check for clarity (and revised accordingly).

Reviewer 3 Report

To the authors,

Congratulations on your interesting research article "Is it safe to buy medications from an app? Assessing medication supply and prescribing apps available in Australia".

Pharmacists play an important role in providing medicine and medicine information to patients. However, in our modern world, the popularity of apps is also influencing the way patients receive their medications in the future and even today. 

This article reviews the supply and prescribing apps available in Australia and assesses their compliance with the Australian National Prescribing Service Medicine Wise.

These apps can compromise patient safety and the safe prescribing of medicines, so it is vital to raise awareness of this important issues.

The manuscript reads very well and the study is well structured and comprehensive. 

Main points:

1)    Legal framework: Line 37 What is the legal framework for online ordering of prescription medicines without in-person (or online) doctor consultation and home delivery of prescription medicines in Australia? It is not entirely clear to me whether both self-selection of prescription medicines and delivery of prescription medicines are legal in Australia. It is not for example not legal in several countries in the EU, so it might be useful to clarify the legal framework with 1-2 sentences in the introduction. 

2)    Quality of medicines: Pharmacists are responsible for ensuring the quality of medicines. Is there a list of legal medicine suppliers/online pharmacies in Australia? Please consider adding possible medicine quality issues as another risk factor of online supply services. There is a risk that medicines sold through unregistered websites may be substandard or falsified. These medicines can also enter the online supply chain more easily. (Only fyi - regulation for buying medicines online in the EU https://www.ema.europa.eu/en/human-regulatory/overview/public-health-threats/falsified-medicines/buying-medicines-online)

3)    Home delivery: The selected medicine is delivered to the patient's home (e.g. line 36). I assume that this means that the medicine is delivered by post and there is no assurance that the medicine is delivered by a courier to the right person and not, for example, to a neighbour (sensitive information) or opened by a child (risk of intoxication). Please consider mentioning this as another risk factor by comparing the delivery of medicines with the "handing over" to the patient in a pharmacy in the discussion.

Minor points:

1)    Lines 173-175: “For these competencies, they were considered to have been met if the app satisfied at least 50% of the developed criteria for each medication.” Please consider mentioning in the discussion that if only 50% of the developed criteria are met, it can be challenged whether the competency was actually sufficiently met.

2)    Lines 296: Can you please add that ‘treatment monitoring is ‘competency 12’

3)    Line 452: Can you please indicate what kind of identification is required by the app? Is it necessary to provide passport number?

4)    Supplementary Table S2-6: Can you please indicate in a legend what “0” and “1” mean, e.g. “0 = not fulfilled”, “1 = fulfilled”?

5)    Table S2: Can you please round the percentages of ‘Medical History’ to whole numbers as in Table S3-6, i.e. 45% instead of 45.45%

Author Response

Dear Reviewer,

Thank you for taking the time to provide such a thorough review of our work. In addition to addressing your comments below, we have re-read the entire manuscript (and amended accordingly) to improve clarity.\

  1. Legal framework: Line 37 What is the legal framework for online ordering of prescription medicines without in-person (or online) doctor consultation and home delivery of prescription medicines in Australia? It is not entirely clear to me whether both self-selection of prescription medicines and delivery of prescription medicines are legal in Australia. It is not for example not legal in several countries in the EU, so it might be useful to clarify the legal framework with 1-2 sentences in the introduction.

Thank you for your comment. At present, there is neither legislation nor regulation in Australia in relation to what constitutes an online (telehealth) consultation. As such, this could potentially encapsulate asynchronous communication (such as completing a web form). Given that all the apps we reviewed mention that prescriptions are authored by Australian-registered doctors, and dispensed by approved Australian pharmacies, the apps, at least on paper, meet Australian legislation. Further, there is currently no legislation restricting home deliveries, provided an appropriate courier is utilized. We have expanded our introduction accordingly.

  1. Quality of medicines: Pharmacists are responsible for ensuring the quality of medicines. Is there a list of legal medicine suppliers/online pharmacies in Australia? Please consider adding possible medicine quality issues as another risk factor of online supply services. There is a risk that medicines sold through unregistered websites may be substandard or falsified. These medicines can also enter the online supply chain more easily. (Only fyi - regulation for buying medicines online in the EU https://www.ema.europa.eu/en/human-regulatory/overview/public-health-threats/falsified-medicines/buying-medicines-online)

Online pharmacies are beyond the scope of the present study. Despite being able to access medications online via the prescribing/supply apps we have identified, the medications are dispensed by approved Australian pharmacies. Thus, issues such as counterfeiting etc are a non-issue in this context. We have realized that there were instances within our manuscript where we used the term online pharmacy. We have reviewed the entire manuscript and updated the wording to avoid confusion.

  1. Home delivery: The selected medicine is delivered to the patient's home (e.g. line 36). I assume that this means that the medicine is delivered by post and there is no assurance that the medicine is delivered by a courier to the right person and not, for example, to a neighbour (sensitive information) or opened by a child (risk of intoxication). Please consider mentioning this as another risk factor by comparing the delivery of medicines with the "handing over" to the patient in a pharmacy in the discussion.

Great comment! We have updated our discussion accordingly.

Minor points:

  1. Lines 173-175: “For these competencies, they were considered to have been met if the app satisfied at least 50% of the developed criteria for each medication.” Please consider mentioning in the discussion that if only 50% of the developed criteria are met, it can be challenged whether the competency was actually sufficiently met.

Thank you for the suggestion. We have updated the discussion accordingly

  1. Lines 296: Can you please add that ‘treatment monitoring is ‘competency 12’

Amended.

  1. Line 452: Can you please indicate what kind of identification is required by the app? Is it necessary to provide passport number?

Amended.

  1. Supplementary Table S2-6: Can you please indicate in a legend what “0” and “1” mean, e.g. “0 = not fulfilled”, “1 = fulfilled”?

Amended.

  1. Table S2: Can you please round the percentages of ‘Medical History’ to whole numbers as in Table S3-6, i.e. 45% instead of 45.45%.

Amended.

Round 2

Reviewer 1 Report

I appreciate the revisions and additions to this interesting paper which I believe improve its relevance to international readers. It might be beneficial to include a link to the Medical Board of Australia's Guidelines for technology-based consultations in the introduction (for the benefit of those in other jurisdictions).

Congratulations on this interesting work.